



# Microwave radiometer observations of the ozone diurnal cycle and its short-term variability over Switzerland

Eric Sauvageat[1,2], Klemens Hocke[1,2], Eliane Maillard Barras[3], Shengyi Hou[1], Quentin Errera[4], Alexander Haefele[3], and Axel Murk[1,2]

[1]Institute of Applied Physics, University of Bern, Bern, Switzerland
[2]Oeschger Centre for Climate Change Research, University of Bern, Bern, Switzerland
[3]Federal Office of Meteorology and Climatology MeteoSwiss, Payerne, Switzerland
[4]Royal Belgian Institute for Space Aeronomy, BIRA-IASB, Brussels, Belgium

**Correspondence:** Eric Sauvageat (eric.sauvageat@unibe.ch)

**Abstract.** In Switzerland, two ground-based ozone microwave radiometers are operated in the vicinity of each other (ca. 40 km): GROMOS in Bern (Institute of Applied Physics) and SOMORA in Payerne (MeteoSwiss). Recently, their calibration and retrieval algorithms have been fully harmonized and updated time series are now available since 2009. Using these harmonized ozone time series, we investigate and cross-validate the strato-mesospheric ozone diurnal cycle derived from the two instruments and compare it with various model-based datasets: a dedicated GDOC ozone diurnal cycle climatology based on the Goddard Earth Observing System (GEOS-5) general circulation model, the Belgian Assimilation System for Chemical ObsErvations (BASCOE) a chemical transport model driven by ERA5 dynamics, and a set of free-running simulations from the Whole Atmosphere Community Climate Model (WACCM). Overall, the two instruments show very similar ozone diurnal cycles at all seasons and pressure levels and the models compare well with each other. There is a good agreement between the models and the measurements at most seasons and pressure levels and the largest discrepancies can be explained by the limited vertical resolution of the microwave radiometers. However, as in a similar study over Mauna Loa, some discrepancies remain near the stratopause, at the transition region between ozone daytime accumulation and depletion. We report similar delays in the onset of the modelled ozone diurnal depletion in the lower mesosphere.

Using the newly harmonized time series of GROMOS and SOMORA radiometers, we present the first observations of short-term (sub-monthly) ozone diurnal cycle variability at mid-latitudes. The short-term variability is observed in the upper stratosphere during wintertime, when the mean monthly cycle has a small amplitude and when the dynamics is more important. It is shown in the form of strong enhancements of the diurnal cycle, reaching up to $4-5$ times the amplitude of the mean monthly cycle. We show that BASCOE is able to capture some of these events and present a case study of one such event following the minor sudden stratospheric warming of January 2015. Our analysis of this event supports the conclusions of a previous modelling study, attributing regional variability of the ozone diurnal cycle to regional anomalies in nitrogen oxide ($NO_x$) concentrations. However, we also find period with enhanced diurnal cycle that do not show much change in $NO_x$ and where other processes might be dominant (e.g. atmospheric tides). Given its importance, we believe that the short-term variability of the ozone diurnal cycle should be further investigated over the globe, for instance using the BASCOE model.



## 1 Introduction

Beyond its role in the earth protection from harmful ultraviolet radiation, ozone is a key species in the energy balance of the middle atmosphere, strongly influencing the radiation budget and thermal state of the stratosphere and mesosphere. Following the success of the Montreal Protocol in 1987, full recovery of the ozone layer is expected for the 21st century with significant regional variability and uncertainties. In particular, there is a high degree of uncertainty about the lower stratospheric ozone recovery and there are increasing observational evidence that ozone is still declining at some locations in the lower stratosphere

(Ball et al., 2018; Maillard Barras et al., 2022a; Godin-Beekmann et al., 2022), without satisfactory explanation to date.

Ozone is a very reactive molecule involved in many (photo-)chemical reactions in the middle atmosphere. The set of pure oxygen photochemical reactions leading to the production and destruction of ozone were first described by Chapman (1930) and are known as the Chapman cycle. Together with catalytic depletion cycles involving many different species ($NO_y$, $Cl_y$, $HO_y$, ...), they mostly drive the ozone amount in the middle atmosphere at multiple time scales. In particular, ozone concentration

are subject to complex diurnal cycle patterns depending on the geographic location, altitude, season and other factors (Schanz et al., 2014), which makes it both important and difficult to fully take into account in models or observations.

The importance of the ozone diurnal cycle in the mesosphere has been recognized early through the use of photochemical models and from early measurements (e.g. Prather, 1981; Vaughan, 1982; Pallister and Tuck, 1983; Zommerfelds et al., 1989). Its main patterns are well known and have been successfully observed and modelled (Connor et al., 1994; Ricaud et al., 1991;

Huang et al., 1997). In the stratosphere though, the ozone diurnal cycle is much weaker which makes accurate observations challenging. However, it needs to be taken into account when comparing different observing systems or to compute accurate trends (Bhartia et al., 2013; Maillard Barras et al., 2020). In recent years, there has been a renewed interest to improve the consideration of ozone and nitrogen oxides diurnal variations in satellite measurements (Frith et al., 2020; Schanz et al., 2021; Strode et al., 2022), partly because of the remaining uncertainties on stratospheric ozone post-2000 trends. For instance, Frith

et al. (2020) used a modified version of the GEOS-5 model to produce a global, zonally averaged ozone diurnal climatology: the GEOS-GMI Diurnal Ozone Climatology (GDOC). The idea was to publish an easy-to-use climatology of scaling factors to account for ozone diurnal variability in intercomparison studies or for the creation of merged ozone dataset. More recently, Strode et al. (2022) develops similar year-specific scaling factors for comparisons with the SAGE III/ISS measurements.

To validate such diurnal scaling factors, accurate observations are needed at different altitudes and locations. However, such

observations remain relatively sparse and challenging, especially in the stratosphere where the ozone diurnal cycle amplitude is small. Also, many satellites are sun-synchronous or use the sun as source which limits their ability to derive full diurnal cycle. Some satellite-based ozone diurnal cycle have been successfully derived however, from SAGE/ISS (Sakazaki et al., 2013, 2015) and from SABER on the Thermosphere Ionosphere Mesosphere Energetics and Dynamics (TIMED) satellite (Huang et al., 2010). Whereas the satellite-based observations offer a global view on the ozone diurnal cycle, they do need to

aggregate data in space or time to derive diurnal cycle, therefore blurring any short-term or regional fluctuations in the cycle amplitude.



Passive microwave ground-based radiometers (MWRs) are well suited for ozone diurnal cycle observations because they operate continuously and do not use the sun as a source. These instruments have been used successfully by different groups to monitor the diurnal cycle, not only in the mesosphere but also in the stratosphere (Connor et al., 1994; Schneider et al., 2005; Haefele et al., 2008; Parrish et al., 2014; Studer et al., 2014; Schranz et al., 2018). In the tropics, Parrish et al. (2014) derived detailed stratospheric ozone diurnal cycle over Mauna Loa from MWR measurements and compared it with satellite measurements and the Goddard Earth Observing System Chemistry Climate Model (GEOSCCM). They found a good agreement between the MWR and satellite observations and remaining discrepancies with the model in the upper stratosphere (3.2 to 1.8 hPa). In the polar region, Schranz et al. (2018) found larger discrepancies between MWR measurements and SD-WACCM simulations over Ny-Ålesund but only focused on a single year of measurements though.

In this contribution, we derive updated ozone diurnal cycles over Switzerland from two collocated (ca. 40 km) ground-based MWRs between 2010 and 2022. The time series have recently been fully reprocessed with a harmonized algorithm (Sauvageat et al., 2022b) and provide a unique set of measurements to study the ozone diurnal cycle and validate model simulations over the mid-latitudes. Compared to the three previous studies on ozone diurnal cycle over Switzerland (Zommerfelds et al., 1989; Haefele et al., 2008; Studer et al., 2014), this study combines the two MWRs over an extended time period ($> 10$ years) and focuses on the time where the two instruments used the same digital spectrometer. The combination of the spectrometer update and of the recent harmonization extended the altitude range and improved the sensitivity of the ozone retrievals. We obtain significant improvements in the updated stratospheric ozone diurnal cycle of GROMOS, which was strongly overestimated compared to the model simulations in the last study (Studer et al., 2014). Given that this instrument provided one of the main reference for ozone diurnal cycle comparison studies over the mid-latitudes in the last decade, we believe that it is highly valuable to present in details these updated results.

In fact, the objective of the present study is multiple. First we use the harmonized time series to derive updated diurnal cycle above Switzerland and provide a comparative basis for different models at mid-latitude. Especially, we aim at providing an additional validation for the dedicated GEOS-GMI Diurnal Ozone Climatology (GDOC), which has been published for use as data analysis tool. In addition, we compare our measurements with two other types of model-based datasets: the Belgian Assimilation System for Chemical Observations from Envisat (BASCOE) chemistry-transport model (CTM) and the Whole Atmosphere Community Climate Model (WACCM) chemistry-climate model (CCM). Finally, we present the first observations of short-term (sub-monthly) ozone diurnal variability and investigate the causes for such variations. We use the global, high-resolution simulations of BASCOE coupled with reanalysis data from ERA5 (Hersbach et al., 2020) to cross-validate our observations. We discuss a case study of the winter 2014-2015 and provide other examples where short-term fluctuations of the ozone diurnal cycle was observed.

The publication is organised as follow: Sect. 2 introduces the datasets and the methods used to compute the ozone diurnal cycle. Section 3 presents the results including the monthly ozone profile comparisons (Sect. 3.1), the intercomparisons of the monthly averaged ozone diurnal cycle over Bern (Sect. 3.2), and an example of observed short-term variability during the boreal winter 2014-2015 (Sect. 3.3). Finally, Sect. 4 presents a summary of the main results and some conclusions.





## 2  Materials and Methods

In the following, we present succinctly the datasets and the methods used in our study. Regarding the datasets, we focus mostly on the microwave radiometer time series and on the main model characteristics, and provide references to the reader for additional details. Also, we summarise the most important features and relevant publications for each dataset in Table 1.

### 2.1  Microwave ground-based radiometers

Microwave ground-based radiometers (MWRs) are passive remote sensing instruments that can be used to derive trace gas or temperature profiles in the atmosphere. MWRs measure the emission of atmospheric molecules in the microwave frequency range. Therefore, they do not rely on the sun for their observations while providing quite high temporal resolution and continuous sampling, which makes them excellent candidates for diurnal cycle studies.

In Switzerland, two ozone microwave radiometers are operated close to each other (ca. $40$ km) on the Swiss Plateau. The GROund-based Millimeter-wave Ozone Spectrometer (GROMOS) is operated by the Institute of Applied Physics (IAP) at the University of Bern ($46.95°$ N, $7.44°$ E, $560$ masl) since 1994 and the Stratospheric Ozone MOnitoring RAdiometer (SOMORA) is operated by the Federal Office of Meteorology and Climatology MeteoSwiss in Payerne ($46.82°$ N, $6.94°$ E, $491$ masl) since 2000. The two instruments have been designed at the IAP, have similar design and use the rotational ozone emission line at $142.175$ GHz to derive strato-mesospheric ozone profiles. Also, they have similar viewing geometries, both observe the sky at $\sim 40°$ elevation angle and experience similar atmospheric opacity conditions. Following discrepancies identified between the two instruments (Bernet et al., 2019; Petropavlovskikh et al., 2019), a complete harmonization of the data processing has recently been performed for GROMOS and SOMORA. It resulted in harmonized, continuous, hourly time series of strato-mesospheric ozone starting in 2009 and which are now freely available (Sauvageat et al., 2022b).

The vertical resolution of the MWRs is quite coarse ($\sim 10$ km up to 3 hPa and $\sim 15$ km above) and the vertical extent of the ozone profile is from 60 to $0.02$ hPa ($\sim 20$ to $75$ km), corresponding to the range where the a-priori contribution to the retrieved profile is lower than $20$ % (Fig. 1). The MWRs coarser vertical resolution needs to be taken into account for intercomparison with higher resolution datasets (e.g. models), also for the diurnal cycle comparisons. The usual way is to apply a smoothing procedure to the higher resolution dataset for the comparisons. In our study, we use the classical "averaging kernel smoothing" which essentially convolve the high resolution dataset with the averaging kernels (AVKs) of the MWR retrieval using Equation 1 (Rodgers and Connor, 2003). Equation 1 also applies the effect of the a-priori contribution of the MWR retrievals onto the higher resolved profile and is usually expressed as:

$$x_c = x_a + \mathbf{A}(x - x_a) \tag{1}$$

with $x_a$ the a-priori profile (derived from monthly WACCM profiles in our case), $A$ the averaging kernel matrix, $x$ and $x_c$ respectively the original and convolved high resolution profile.

GROMOS and SOMORA essentially have the same sensitivity, allowing to compare their observations directly. It can be seen by looking at the mean AVKs and the measurement contribution of the retrievals shown in Fig. 1. It also means that





there is only little difference whether we use GROMOS or SOMORA AVKs for the smoothing procedure. In the following, all convolutions on the higher resolution profiles are performed using the GROMOS AVKs. Figure 1 shows the mean AVKs

of the full GROMOS and SOMORA time series, however, for all the convolutions we use the appropriate monthly daytime or nighttime AVKs.

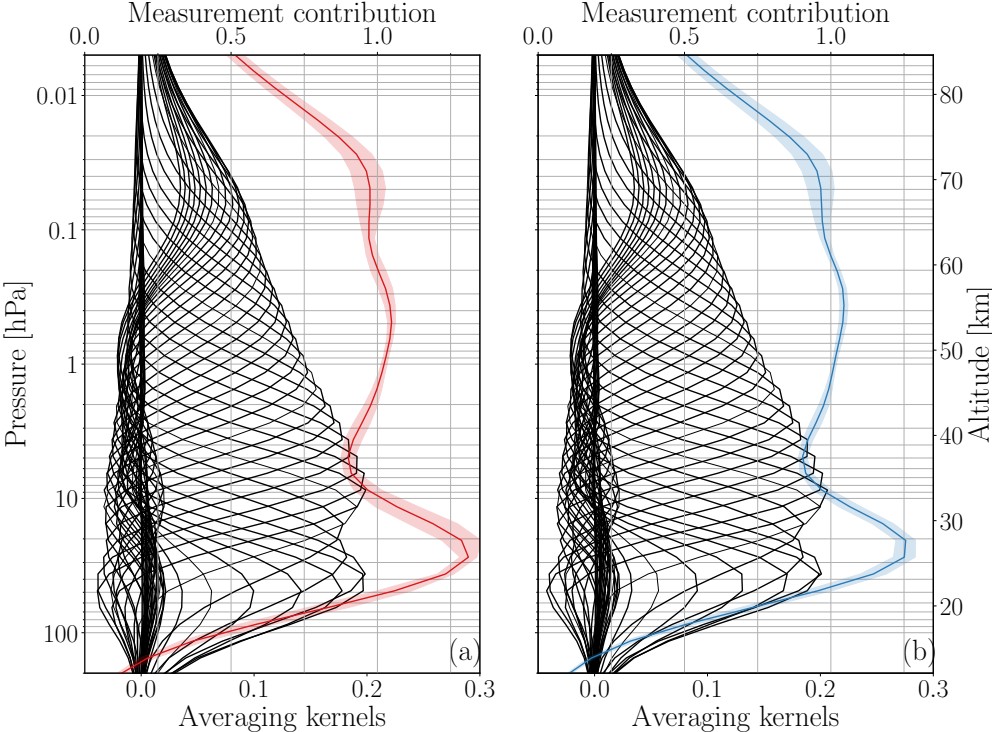

**Figure 1.** Mean averaging kernels and measurement contribution for (**a**) GROMOS and (**b**) SOMORA. The black lines are the averaging kernel at individual pressure levels whereas the color lines are the respective measurement contribution (see upper x-axis). The shaded color area shows the standard deviation of the measurement contribution.

### 2.1.1 Satellite measurements

For validation purpose, we use measurements from the Microwave Limb Sounder (MLS) mounted on the Aura spacecraft (Waters et al., 2006). Since its launch in July 2004, the MLS instrument has been used extensively for trace gas observations

and is one of the main measurement reference for global ozone monitoring studies. More specifically, we use the latest ozone retrieval product (v5), following the screening guidelines provided in Livesey et al. (2022). The MLS ozone vertical resolution ranges from $\sim 2.5$ to $\sim 5$ km in the stratosphere and mesosphere whereas its horizontal resolution ranges between 300 and 500 km. As spatial collocation criteria, we keep only measurements around Switzerland ($\pm 1.8°$ in latitude and $\pm 5°$ longitude). Aura overpasses Switzerland twice a day, at 02:20 LST and 13:10 LST, thus providing the day-to-night ozone ratios but not



the full diurnal cycle. In Sect. 3.3, we also show some measurements of temperature and nitrous oxide from MLS, however, as higher temporal resolution was needed for the short-term analysis, these were obtained with more relaxed collocation criteria ($\pm 3.6°$ latitude and $\pm 10°$ longitude) but following the same screening guidelines (Livesey et al., 2022)

## 2.2 Model-based datasets

### 2.2.1 GDOC

The GEOS-GMI Diurnal Ozone Climatology (GDOC) is a model-based climatology of ozone diurnal cycle derived from the NASA Goddard Earth Observing System general circulation model, version 5 (GEOS-5). The goal of this climatology is to provide a simple data analysis tool to account for ozone diurnal variability, e.g. when comparing different satellites profiles.

For the production, GEOS-5 was run in replay mode constrained to 3-hourly MERRA-2 assimilated meteorological fields from January 2017 to December 2018 (see Orbe et al. (2017); Frith et al. (2020) and references therein for model details). As

final product, the GDOC provides zonally averaged (on $5°$ latitude bands) ozone diurnal cycle from 90 to $0.3$ hPa ($\sim 20$ to $55$ km) with equivalent vertical resolution of $\sim 1$ km and a time resolution of 30 minutes. The climatology is also available (on request) on original model levels but has not been evaluated below 30 and above $0.3$ hPa, which is the reason why we chose not to use it outside of this pressure range.

It provides monthly climatological ozone values as a function of local solar time (LST) normalized by midnight ozone values.

The GDOC does not contain the original ozone profiles which prevents the application of the averaging kernel smoothing procedure on this dataset. Consequently, we only show the original high resolution profile from the GDOC dataset.

### 2.2.2 BASCOE

The Belgian Assimilation System for Chemical ObsErvations (BASCOE) is a chemistry transport model (CTM) developed at the Royal Belgian Institute for Space Aeronomy (BIRA-IASB) (Errera and Fonteyn, 2001; Errera et al., 2008). For this study,

the model was run globally and without data assimilation from 2010 to 2020 in a similar setup as described in Chabrillat et al. (2018). BASCOE dynamics relies on 6-hourly winds and temperature taken from ERA5, the latest reanalysis dataset from the European Center for Meteorological and Weather Forecast (ECMWF). The model has a time resolution of 30 minutes, and a spatial resolution of $2°$ in latitude and $2.5°$ in longitude. It runs on $42$ hybrid pressure levels from the surface to the mesosphere ($\sim 0.01$ hPa) with a vertical resolution of $\sim 1-4$ km in the stratosphere and lower mesosphere. For this study, the model

outputs have been interpolated at the location of Bern so that they can be compared with our two MWRs.

### 2.2.3 WACCM

We also use results from the Whole Atmosphere Community Climate Model (WACCM), version 4 in the configuration described by Schanz et al. (2021). WACCM is a fully coupled global chemistry-climate model developed at the National Center for Atmospheric Research (NCAR) with a stratospheric chemistry module based on the Model of Ozone and Related Chemical

Tracers (MOZART) (Kinnison et al., 2007). It simulates the atmosphere from the surface to $\sim 150$ km altitude, with vertical



resolution ranging between 1.1 and 2 km in the middle-atmosphere. WACCM was run with a horizontal resolution of 4° latitude by 5° longitude and for our study, we use the closest grid point to both Bern and Payerne, which corresponds to 46° N and 5° E. The model was run with the pre-defined free-running "F 2000" scenario, simulating a perpetual year 2000 but without data nudging.


**Table 1.** Summary of the datasets used in this study.

| Dataset | Type | Coverage (horizontal, vertical) | Reference |
|---------|------|--------------------------------|-----------|
| GROMOS | MWR measurement | local, $60 - 0.02$ hPa | Sauvageat et al. (2022b) |
| SOMORA | MWR measurement | local, $60 - 0.02$ hPa | Sauvageat et al. (2022b) |
| MLS | Limb sounding measurement | global, $261 - 0.001$ hPa | Waters et al. (2006); Froidevaux et al. (2008) |
| GDOC | Model based climatology | zonal, $30 - 0.3$ hPa | Frith et al. (2020) |
| BASCOE | CTM | global, surface$-0.05$ hPa | Errera et al. (2008) |
| WACCM | CCM, free-running | global, surface$-5.1 \cdot 10^{-6}$ hPa | Garcia et al. (2007); Marsh et al. (2013) |
| ERA5 | Reanalysis | global, surface$-0.05$ hPa | Hersbach et al. (2020) |

## 2.3 Ozone profiles and day-to-night ratios

Before computing the ozone diurnal cycle from our different datasets, we first compare their monthly averaged ozone profiles during daytime and nighttime and compute their day-to-night ratios. We compare our two MWRs with MLS and the BASCOE simulation, all averaged on the time period 2010-2020, therefore removing most of the year-to-year variability. WACCM is not
included in these comparisons for two reasons. First, we use the free-running WACCM simulations for a single year, and it would make no sense to compare it with the other datasets multi-year averages, especially in the wintertime when the dynamics is an important modulator of the ozone amount. Second, the monthly averaged WACCM ozone profiles are actually used as a-priori for our MWR retrievals and therefore, they can not be used for validation against the retrieved MWR profiles.

For the ozone profile comparisons, we use the approximate MLS overpass times, keeping all timestamps between 12:00 and
15:00 LST for the daytime profiles and between 01:00 and 04:00 LST for the nighttime profiles, regardless of the dataset. As explained in Sect. 2.1, we convolve the BASCOE simulations and the MLS measurements with monthly averaged daytime or nighttime GROMOS AVKs.





## 2.4 Ozone diurnal cycle

Similarly to the GDOC climatology, we choose to express the ozone diurnal cycle as ratio of ozone VMR relative to the
midnight value. To compute the reference midnight value ($O_{3,NT}$ in Eq. 1), we have used different time periods to reflects the
different time resolution of the datasets. For the MWRs ($\sim 1$ h time resolution), we compute the midnight reference value by
taking an average of 2 nighttime measurements between 23:00 LST and 01:00 LST. For WACCM and BASCOE (30 minutes
time resolution), we use measurement between 00:00 and 01:00 LST, whereas the GDOC was normalized to the values between
23:45 to 00:15 LST Frith et al. (2020).

For each hour and at each pressure level, we then compute the ratios to ozone at midnight using Equation 2. To simplify the
notation, we do not explicitly write the pressure dependence of all the terms.

$$\Delta O_3(h) = \frac{O_3 - O_{3,NT}}{O_{3,NT}} \tag{2}$$

To compare with the monthly GDOC climatology, we compute monthly averaged $\Delta O_3(h)$ from GROMOS, SOMORA,
WACCM and BASCOE. For each dataset, we use the available time series, i.e. 12 years of data for the two MWRs (2010-2022),
10 years for BASCOE (2010-2020), and the 1 year of the WACCM free-running model run. For the MWRs, we additionally
filtered the time series to remove the measurements done at very high tropospheric opacity ($\tau > 1.5$) as they result in lower
quality retrievals and can potentially contaminate the monthly averages. To some extent, it also helps to limit any seasonal bias
arising due to the summertime higher opacity although it is difficult to rule out this effect completely (e.g. see discussion on the
effect of the opacity on GROMOS and SOMORA in Sauvageat et al. (2022b)). For the diurnal cycle, the effect of this filtering
is not very large and for the interested reader, the unfiltered version is provided in the supplementary material (Figures S5 to
S16).

We compute errors on the MWRs and BASCOE ozone diurnal cycles as standard error of the mean (SEM). For each month
and LST hour, we compute the standard deviation of $\Delta O_3(h)$ and divide it by the square root of the number of ozone profiles
available for each hour.

## 2.5 Short-term variability of the ozone diurnal cycle

In addition to monthly averaged ozone diurnal cycle, we also show observations of short-term (sub-monthly) variability of
ozone diurnal cycle. GROMOS and SOMORA provide a unique setup for short-term ozone diurnal cycle observation because
they have continuous, hourly, collocated measurements. Therefore, we can use them to compute the ozone diurnal cycle on
sub-monthly period and cross-validate their measurements. Also, we use BASCOE simulations to compute the short-term
variability of the ozone diurnal cycle over Switzerland and to investigate the cause of such variability.

In order to detect sub-monthly variations in the ozone diurnal cycle, we computed the day-to-night differences in ozone
VMR for GROMOS, SOMORA and BASCOE. More specifically, we compute the anomalies of the day-to-night differences
($D_{O_3}$) to a monthly climatology. As daily anomalies are too noisy, we average these differences on 5 days.



$$D_{O_3} = O_{3,DT} - O_{3,NT} \ [\text{ppmv}] \tag{3}$$

In this contribution, we focus on the winter 2014-2015 and use BASCOE and MLS data to investigate and discuss potential reasons explaining a specific event during this winter. For further studies, we provide along with this publication the full time series of $D_{O_3}$ daily anomalies for GROMOS, SOMORA and BASCOE.

## 3    Results and discussions

### 3.1    Monthly ozone profiles and day-to-night ratio

The comparisons of the monthly averaged ozone daytime and nighttime profiles and day-to-night ratios are shown in Fig. 2 for December and Fig. 3 for June as proxies for the winter and summer season. Figure A1 and Fig. A2 in Appendix A show similar comparisons for March and September respectively. Similar comparisons but with respect to SOMORA MWR can be seen in the supplementary material (Figures S1 to S4).

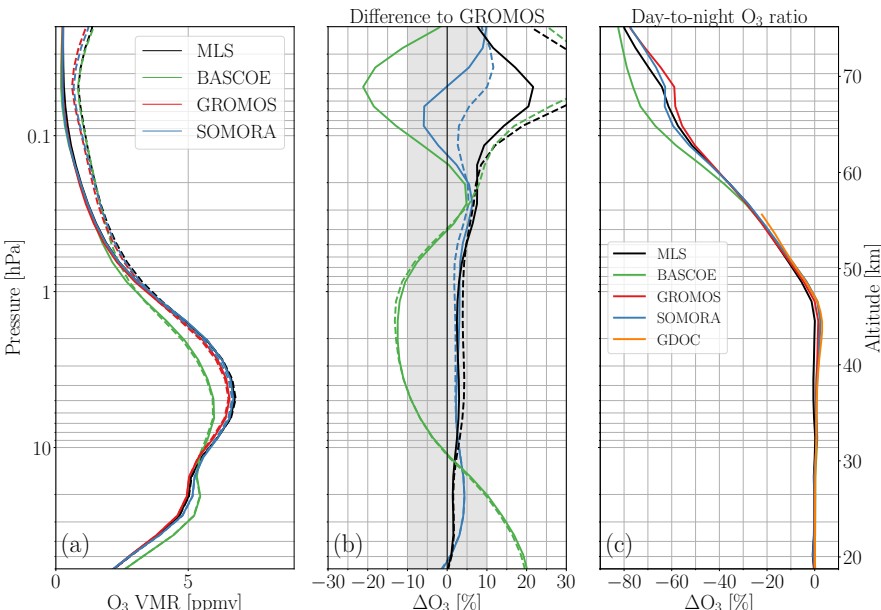

**Figure 2.** (**a**) Monthly averaged profiles, (**b**) relative differences ((X-GRO)/GRO), and (**c**) day-to-night ratios of ozone VMR above Switzerland in December. In (**a**) and (**b**), the solid lines are the daytime profiles whereas the dashed lines are the nighttime profiles.





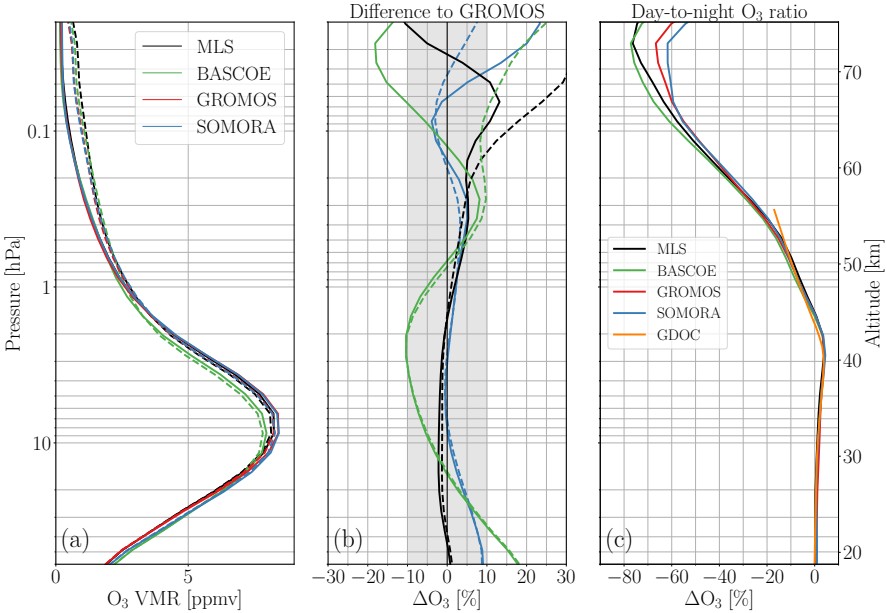

**Figure 3.** Same as Fig. 2 but for June.

Overall we find a good agreement between measured ozone profiles (GROMOS, SOMORA and MLS), with relative differ-
ences between the measured ozone profiles lower than 10 % up to 0.1 hPa. The differences between BASCOE and the MWR
are within 15 % between 30 and 0.2 hPa, with slightly bigger bias during the winter months. BASCOE notably underesti-
mates ozone amounts in the upper stratosphere to lower mesosphere (up to $\sim 0.5$ hPa) and overestimates ozone in the lower
stratosphere, regardless of the month or the time of day. Above 0.2 hPa, the differences depend on the month considered but
the model tends to underestimate the daytime ozone profiles, leading to a small overestimation of the day-to-night depletion
ratio over 0.2 hPa. The ozone deficit of BASCOE in the middle atmosphere is consistent with previous studies (Skachko et al.,
2016) and is a bias found in other models as well (see e.g. Fig. 3.1 in SPARC, 2010). One reason for this ozone deficit might
be an overestimation of $NO_2$ in the model simulations, enhancing catalytic ozone destruction though the so-called "Crutzen"
cycle (Crutzen, 1970).

All datasets show the ozone daytime accumulation in the stratosphere and its transition to ozone daytime depletion around
the stratopause ($\sim 2 - 1$ hPa). All datasets show stratospheric ozone accumulation during daytime (up to $\sim 1$ hPa) with
stronger amplitude during the summertime, and they all show the strong mesospheric ozone depletion during daytime, growing
in amplitude with altitude. They agree quite well on the pressure at which peak accumulation and peak depletion occur. During
summer, there is an excellent agreement between day-to-night ratios up $\sim 0.6$ hPa. Above this pressure level, the GDOC
climatology systematically underestimates the daytime ozone loss, leading to less negative day-to-night ratio than the other
datasets. In the mesosphere, BASCOE and MLS compare well with the MWRs up to 0.05 hPa, with day-to-night ratios in
agreement within 15 %.





## 3.2 Comparison of monthly ozone diurnal cycle

Going beyond the day-to-night ratio, full monthly diurnal cycles over Switzerland are shown for summer and winter in Figures 4 and 5 for the 2 MWRs and the 3 model-based datasets (Figures B1 and B2 in Appendix B show similar results for spring and
autumn respectively). These figures show the ratios to ozone midnight values as a function of the LST between $60$ and $0.02\,\mathrm{hPa}$ (respectively $30$ and $0.3\,\mathrm{hPa}$ for GDOC). As mentioned previously, the monthly averages correspond to different time periods for each dataset: 2010-2022 for GROMOS and SOMORA, 2010-2020 for BASCOE, 2017-2018 for the GDOC and 2000 perpetual year for WACCM. Also, AVK smoothing procedures have been applied to the WACCM and BASCOE dataset but not to the GDOC. The original version without any AVK smoothing can be seen for all datasets in the supplementary material.
For better visualization, we also show these diurnal cycles averaged over 9 selected pressure ranges. This is shown in Figures 6 and 7 for winter and summer (in Figures B3 and B4 in Appendix B for spring and autumn). These figures show the original cycle from each dataset together with the AVK smoothed cycle, which enables to clearly see the effect of the AVK smoothing procedure on the high-resolution datasets.

The new harmonized ozone time series from GROMOS and SOMORA have excellent agreement in ozone diurnal cycle.
They agree well in patterns and amplitudes at all seasons and most altitudes. Some small discrepancies can be seen in summertime in the transition region, however, as shown in Sauvageat et al. (2022b), it is also the season where GROMOS and SOMORA experienced the larger discrepancies between their respective measurements. The two regimes of the ozone diurnal cycle are clearly visible in all datasets. Namely, the accumulation of ozone during daytime in the stratosphere and the depletion of ozone during daytime above $\sim 1\,\mathrm{hPa}$ are well captured by all datasets.

Among the model datasets, we observe most discrepancies of the diurnal cycle amplitude by the GDOC during wintertime in the upper stratosphere (see also the month of January and February, shown in the supplementary material). To some extent, these discrepancies could be due to the temporal (different averaging periods) and longitudinal (zonal mean in GDOC) variability, which are both smoothed out in the GDOC. As mentioned by Frith et al. (2020), this is also the season where the ozone diurnal cycle is smaller and where the model uncertainties are higher. Below, we will present a summary of the differences between
the MWRs and the models focusing on different altitude regions and discuss in more details the reasons for the observed discrepancies.





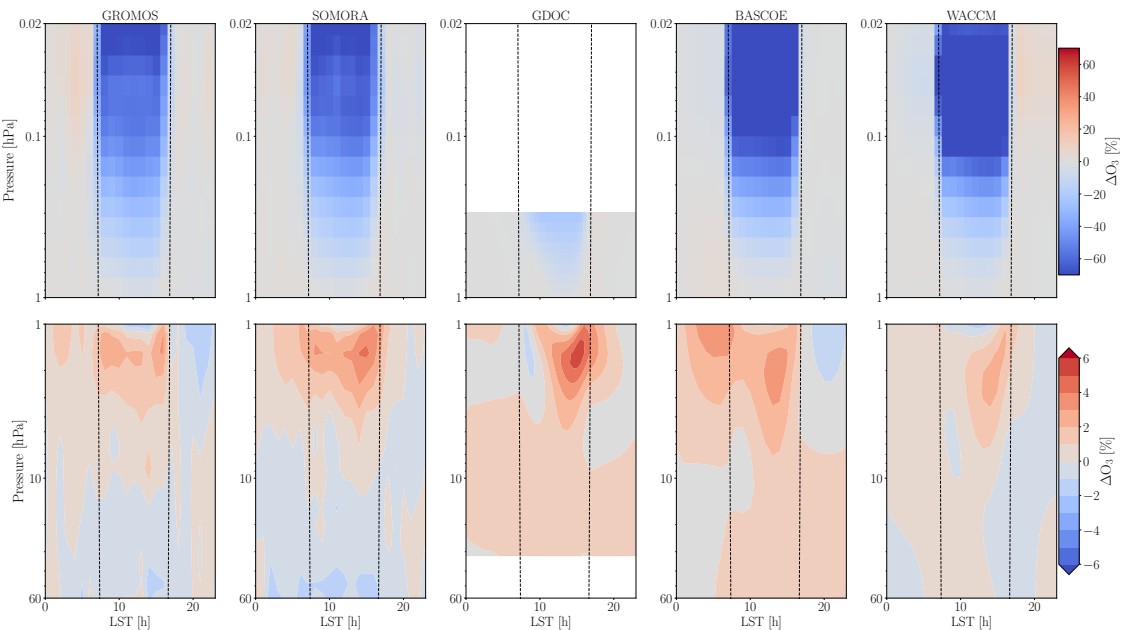

**Figure 4.** Monthly averaged ozone diurnal cycle over Switzerland in December as seen in GROMOS, SOMORA, GDOC, BASCOE and WACCM datasets. Note that only the BASCOE and WACCM datasets have been convolved with the AVKs of GROMOS as explained in Sect. 2.1.

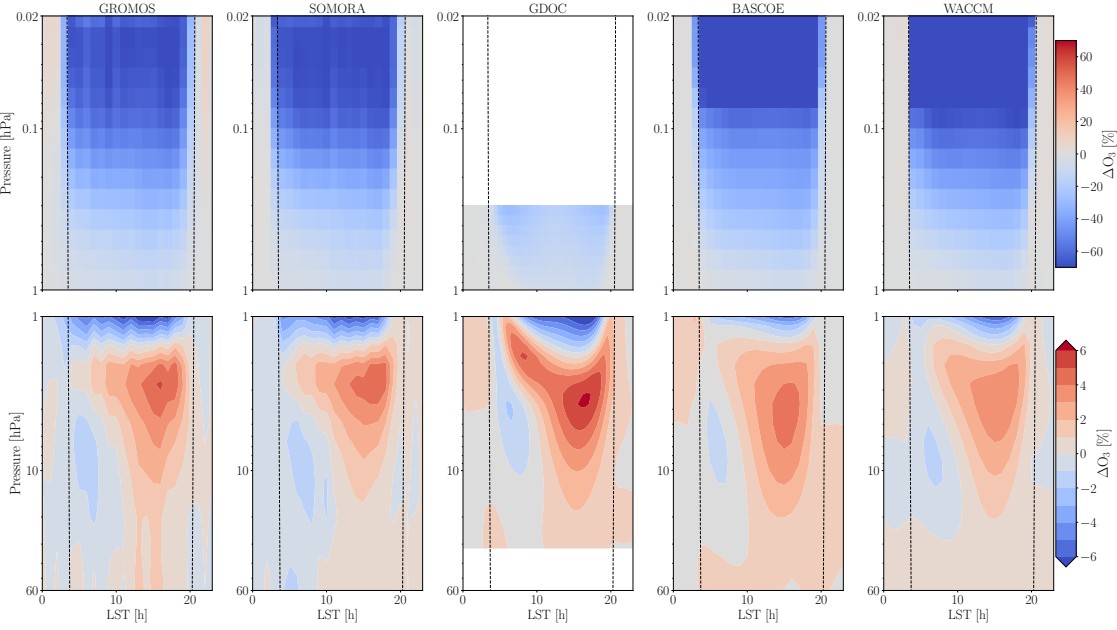

**Figure 5.** Same as Fig. 4 but for June.



**Figure 6.** Monthly averaged ozone diurnal cycle over Switzerland at 4 selected pressure ranges in December. For BASCOE and WACCM we show both the diurnal cycle before (solid lines) and after convolution (dash lines) of the dataset with the AVKs of GROMOS. To account for the large changes of the diurnal cycle amplitude with altitude, the scale of the y-axis is adapted for each sub-plot.







**Figure 7.** Same as Fig. 6 but for June.

### 3.2.1 Mesosphere (p < 0.3 hPa)

Overall, we observe a tendency of the models to overestimate the diurnal ozone depletion in the mesosphere. It is mostly noticeable above $\sim 0.1$ hPa where the sensitivity of the MWRs are decreasing and the measurement error is growing fast.

Therefore, even if the effect of the lower sensitivity should be included through the AVK smoothing, biases above this altitude should be considered with care. Note that at this altitude, BASCOE also has a limited vertical resolution as it only uses 2 pressure levels above 0.1 hPa. Considering the above limitations, we still observe quite a good agreement of the upper



mesospheric diurnal cycle at all seasons. In agreement with Parrish et al. (2014) but in contradiction with the conclusions from Studer et al. (2014), we do not observe a significant seasonal variation of the mesospheric diurnal cycle amplitude. This is in

better agreement with the model results, which show similar amplitude throughout the year.

### 3.2.2 Lower mesosphere (1 − 0.3 hPa)

In the lower mesosphere, we note a consistent bias between the models and the observations around sunrise: the diurnal ozone depletion observed by the MWRs consistently starts earlier than the models. It is true for most months and could be partly explained by differences in the vertical resolution (e.g. see Fig. 7 at 51 and 56 km). Interestingly, it does not seem to impact

the sunset period which rules out potential errors arising from the time conversion between the different datasets. This feature was also observed by Parrish et al. (2014) over Mauna Loa, and it seems to persist even after application of the AVKs (not for all months though), gives us confidence that it does not result from the a-priori.

### 3.2.3 Stratopause region (3 − 1 hPa)

Around the stratopause, we can clearly see the complex transition region between the mesospheric diurnal depletion and the

stratospheric accumulation. This is where we notice the largest biases between our different datasets. In fact, we observe discrepancies among the three model-based datasets, and between the observations and the model-based datasets. The bias around the stratopause (1 − 3 hPa) are similar to the ones reported by Parrish et al. (2014) and Haefele et al. (2008), i.e. differing behaviour in the pre-dawn hours and after sunrise. They are seen at all seasons during daytime and reach values up to 2 % differences among the models themselves (e.g. between 1 and 2 hPa in Fig. 7). Between the models and the MWRs, the

biases are significantly reduced by the application of the AVK smoothing procedure but we still note biases up to 2 % in this region.

As will be shown in Sect. 3.3, the upper stratosphere and lower mesosphere are also experiencing short-term variability of the ozone diurnal cycle, which can influence the monthly averaged cycle. In particular, the datasets produced using only a few specific years (i.e. WACCM or GDOC in our case), will be influenced by the short-term variability of these years whereas it

will be smoothed out in the MWR or the BASCOE datasets which are averaged on 10 years or more. As shown in Figure S11 from the supplementary material of Frith et al. (2020), whereas the inter-annual variability is generally limited below 5 hPa and during summer, inter-annual variations up to 5 % around 0.5 − 1 hPa can be seen during wintertime in the mid-latitudes. It supports the existence of short-term variability in the ozone diurnal cycle and might therefore explain some of the remaining discrepancies near the stratopause region.

### 3.2.4 Middle and lower stratosphere (30 − 3 hPa)

In the middle and lower stratosphere, we observe the typical behaviour of the stratospheric ozone diurnal cycle: a small dip after sunrise followed by a gentle accumulation reaching a maximum in the late afternoon. The stratospheric cycle shows a high seasonal variability, with a maximum diurnal cycle amplitude around the summer solstice and lower diurnal variations



during winter. In summer, we observe a peak amplitude of the ozone diurnal cycle of $3 - 4\,\%$ in the afternoon around $5\,\mathrm{hPa}$ in

July, reducing to less than $2\,\%$ in the wintertime. For this reason, the dip after sunrise, attributed to rapid dissociation of $NO_2$ at sunrise (Pallister and Tuck, 1983), is mostly visible during the summer months. Note that this is a significant improvement compared to the previous retrievals of GROMOS time series, where the dip was not observed and where the amplitude of the stratospheric cycle was high compared to the models (Figure 6a and 6b in Studer et al. (2014)). With the new time series, the amplitude of stratospheric ozone cycle is well captured by GROMOS and SOMORA at most seasons. In fact, most of the

discrepancies that we observe in the middle stratosphere are the consequences of the limited vertical resolution of the MWRs, whereas the differences of the lower stratosphere stay mostly within the error bars.

### 3.3 Short-term variability of the ozone diurnal cycle

In this section, we present the first measurements of short-term ozone diurnal cycle variability using the unique setup offered by the collocated, hourly resolved measurements from GROMOS and SOMORA. To our knowledge, it is the first time that

short-term variability of the ozone diurnal cycle is observed and in the following, we try to identify some of the reasons leading to such events focusing on a case study from the boreal winter 2014-2015.

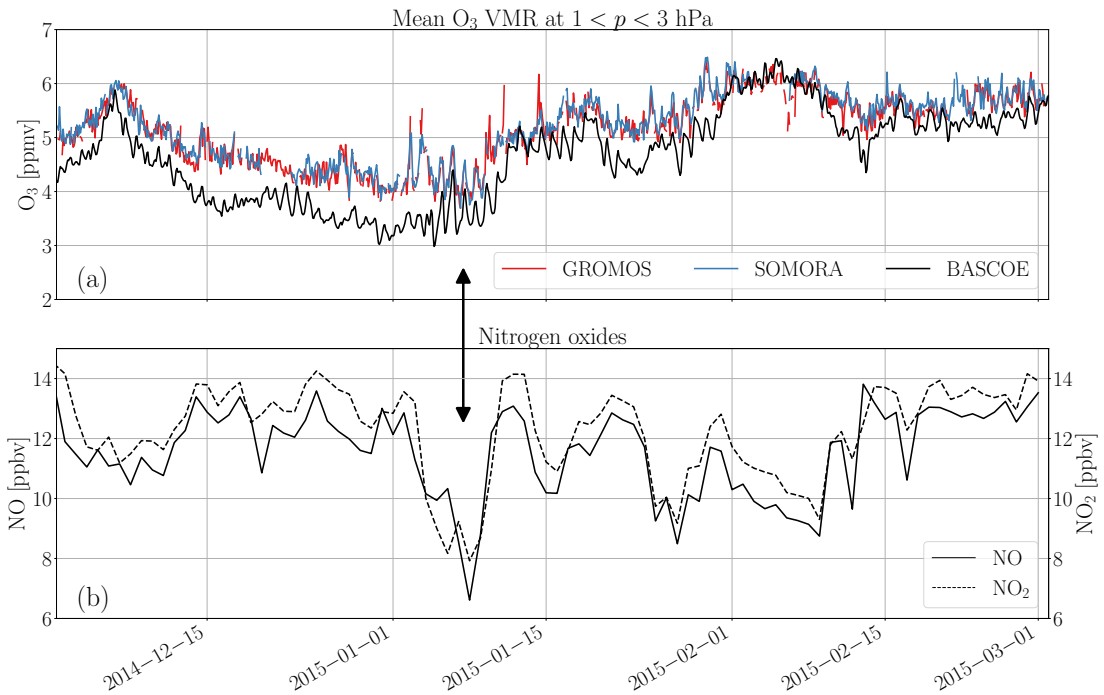

**Figure 8.** (a) Ozone VMR from GROMOS, SOMORA and BASCOE during the boreal winter 2014-2015. (b) Nitrogen oxides (NO and $NO_2$) simulated by BASCOE. All quantities are averaged between 3 and $0.8\,\mathrm{hPa}$ and the ozone time series over $2\,\mathrm{h}$ time periods. $NO_x$ are shown as daily mean of nighttime ($NO_2$) respectively daytime (NO) values. The arrow highlights the period with an enhanced diurnal cycle.



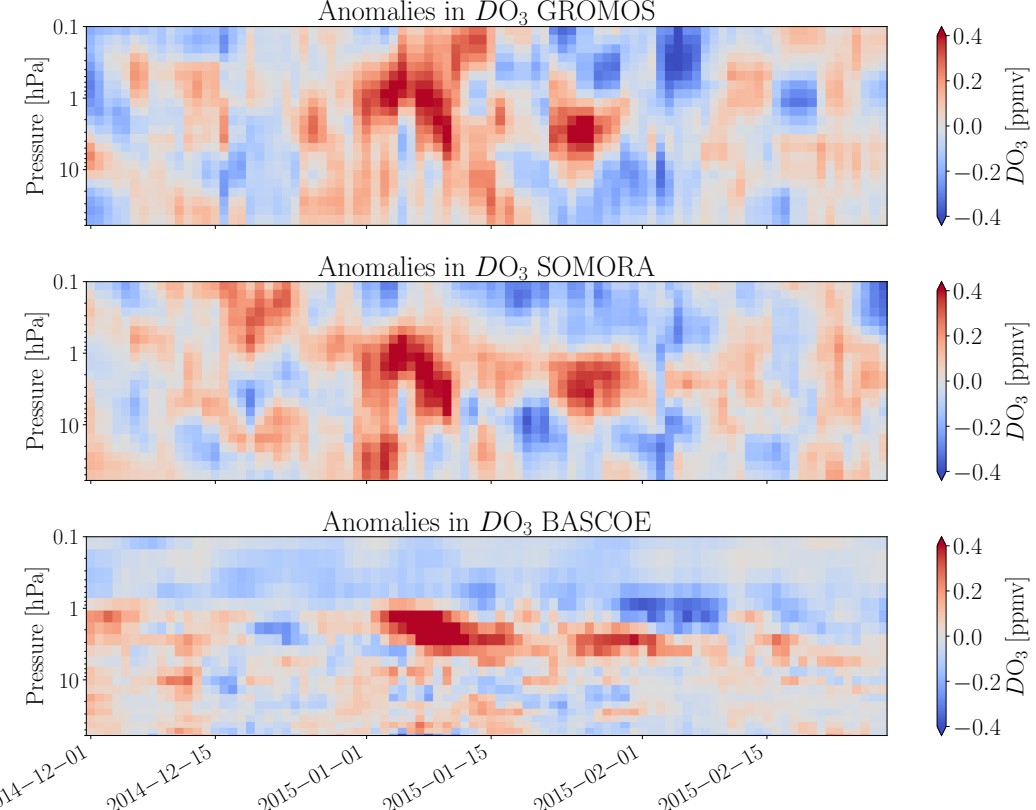

**Figure 9.** Anomalies in day-night $D_{O_3}$ from GROMOS, SOMORA and BASCOE during the boreal winter 2014-2015. For each dataset, we show the differences of $D_{O_3}$ compared to a monthly climatology computed on the decade 2010-2020.

The upper panel in Fig. 8 shows the ozone concentration in the upper stratosphere from GROMOS, SOMORA and BASCOE. From the ozone times series already, there are time periods where the ozone VMR shows some large fluctuations on a diurnal basis in a season where the mean ozone cycle is usually small (see Fig. 6, around 40 km). An example of such case can be seen

at the beginning of January 2015 (arrow on Fig. 8) or, for instance, at the end of January 2015. From the BASCOE time series, we can even identify other period with enhanced cycle which are not really seen as such in the MWRs measurements. These are some examples of what we refer as "short-term ozone diurnal cycle variability", generally lasting for a few days.

These enhancements can be better seen in Fig. 9, in the form of day-to-night $D_{O_3}$ anomalies in the middle atmosphere $(60 - 0.1 \text{ hPa})$. It shows similar patterns in GROMOS and SOMORA time series, with large increase of $D_{O_3}$ around 1 hPa

at the beginning of January 2015, followed by a secondary peak in the second half of the month. To some extent, BASCOE is also able to reproduce these two peaks in the ozone diurnal cycle, somehow limited to below 1 hPa and with limited vertical resolution.





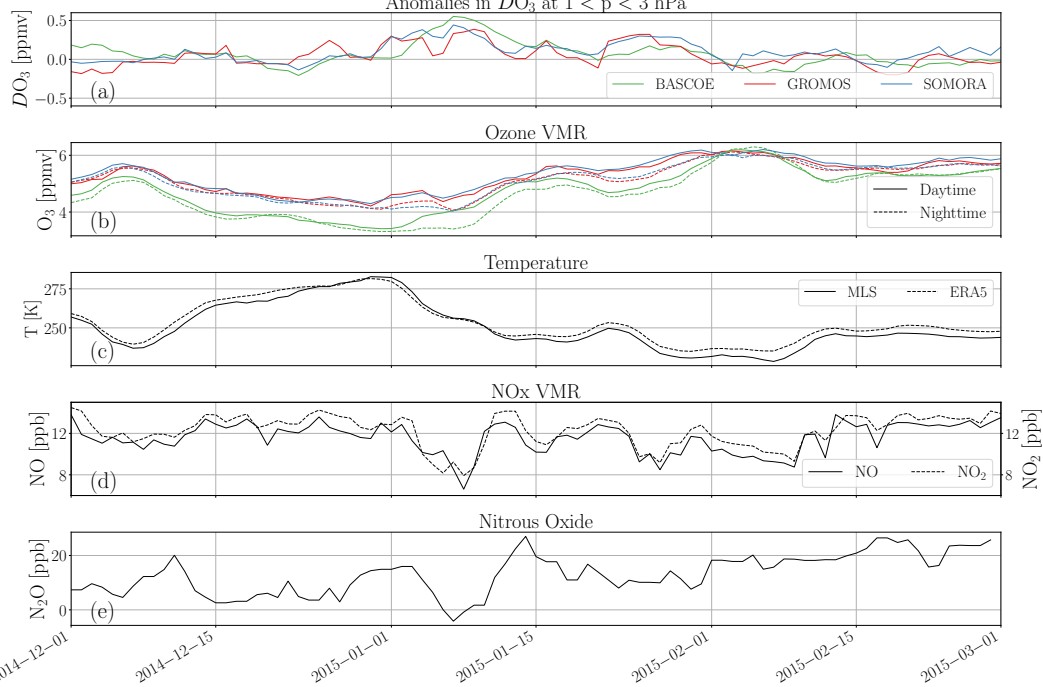

**Figure 10.** Time series of different quantities during the winter 2014-2015, all averaged between 3 and 0.8 hPa. (a) is the $D_{O_3}$ anomalies, (b) shows the ozone VMR of the three dataset during daytime and nighttime, (c) is temperature from MLS and ERA5, (d) are NO and $NO_2$ as simulated by BASCOE and (e) is $N_2O$ measurements from MLS.

Focusing on the upper stratosphere ($3 - 1$ hPa) where the anomalies are highest, Fig. 10 shows the temporal evolution of different quantities during the winter 2014-2015. In particular, Fig. 10(b) shows the day and nighttime ozone values. Focusing
on the early January event, it can be seen that there is a consistent increase in daytime ozone from the three datasets, somehow delayed slightly in time in BASCOE compared to the MWRs. Such increase is also visible during the second peak at the end of January for GROMOS and SOMORA and somehow less clearly in the BASCOE series. In terms of amplitude, this increase is substantial and corresponds approximately to $4$ to $5$ times the monthly averaged day-to-night difference in January which is $\sim 0.1$ ppmv at this pressure level.
Corresponding to these peaks in $D_{O_3}$ anomalies, sharp decreases in nitrogen oxides ($NO_x$) are simulated from BASCOE. In fact, the decrease affects both nitric oxide (NO) and nitrogen dioxide ($NO_2$). The two species are in photochemical balance during the day as NO mainly originates from photolysis of $NO_2$ during daytime and can react with ozone to give back $NO_2$, forming a major catalytic ozone depletion process in the middle atmosphere (Crutzen, 1970). In fact, the effect of $NO_x$ on ozone maximizes between 20 and 45 km altitude ($\sim 50 - 1$ hPa), corresponding well to the peaks of diurnal cycle enhancements
seen on Fig. 9. To cross-validate these simulations, we also show some nitrous oxide ($N_2O$) measurements taken by the MLS instrument onboard the Aura satellite in Fig. 10(e). Also here, the early January peak is visible as a decrease in the





N$_2$O measurement from MLS, which makes sense as N$_2$O is the main source of NO$_x$ in this altitude region (McElroy and McConnell, 1971). Note that this is not the only event identified where such a behaviour can be seen. In fact, it seems that most winters seem to experience similar events (see e.g. similar plots for the boreal winter 2016-2017 shown in Appendix C)

In order to provide a more global picture and investigate the reasons for the N$_2$O decrease seen above Central Europe in the MLS measurements, we investigated the dynamical situation of the Northern Hemisphere by looking at the ERA5 reanalysis data during this period. In fact, this event follows closely a Sudden Stratospheric Warming (SSW) which took place early January. It was a minor warming but with significant disturbances on middle atmospheric chemistry and transport (Manney et al., 2015). In fact, Fig. 2 from Manney et al. shows how the polar vortex briefly splitted at the onset of the SSW, leading to

a mixing of the air between the mid-latitude and the poles in the upper stratosphere. Following this event, some filaments of polar air containing little ozone and N$_2$O reached Central Europe as can be seen on the ozone map in Fig. 11. Such an irruption of polar air over Switzerland would explain the decrease in the N$_2$O MLS measurements seen early January and might well explain the subsequent changes NO$_x$ and consequently in the ozone diurnal cycle.

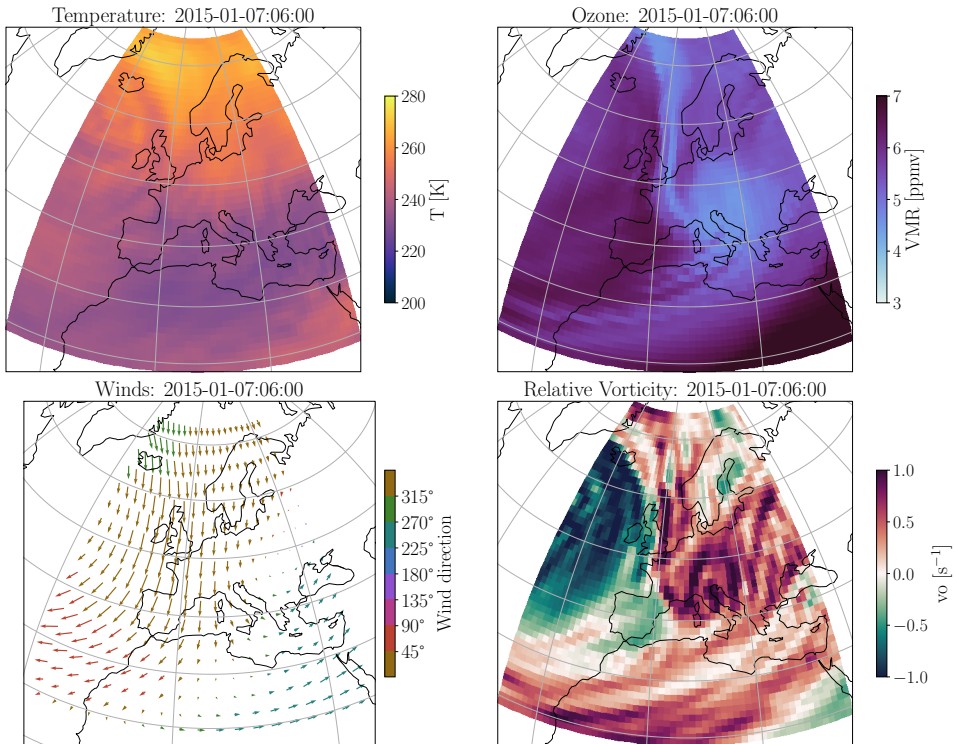

**Figure 11.** Situation over Europe in the upper stratosphere $\sim 5$ hPa shortly after the minor SSW of early January 2015 as seen in the ERA5 reanalysis.

Interestingly, we do not observe the strong ozone decrease associated with this filament of polar air reaching Bern, neither in

the MWRs nor in the BASCOE time series. This could be a problem from the ERA5 reanalysis data as it does not feature any





diurnal cycle at these altitude and therefore might also lack the reaction of ozone to greater sun illumination (resulting in more ozone production and therefore increasing ozone amount in the low-ozone polar air which would be missed by the model). Even though this event might be considered textbook example of such dynamical event, we find it interesting to find such a coherent picture of a short-term event from a combination of ground-based measurements, chemistry transport model, satellite
measurements and reanalysis data.

## 4 Conclusions

Using new harmonized ozone time series from two nearby located microwave radiometers enables us to study in great details the ozone diurnal cycle over Switzerland. With more than 11 years of parallel, independent measurements, these instruments provide a unique validation source for satellite and model-based datasets. We find that the recently published GDOC clima-
tology compares well with our MWRs above Switzerland and agrees well with the WACCM and BASCOE models in the stratosphere and lower mesosphere. As reported by previous studies, we observe some remaining discrepancies between our observations and the models near the stratopause, in the transition region between ozone daytime accumulation and depletion. The discrepancies remain small and are significant only during summertime, where the diurnal cycle is stronger, providing better signal-to-noise ratio for the observations. Some of our results contradict a previous study also based on the GROMOS
instrument (Studer et al., 2014), now providing a better agreement of the ozone diurnal cycle compared to model-based dataset but also compared to other previous MWR diurnal cycle study (Parrish et al., 2014). These updated results motivated the present study, and are a consequence of the spectrometer change and of the recent harmonization of the calibration and retrieval routines of GROMOS and SOMORA (Sauvageat et al., 2022b).

For the first time, short-term variations of the ozone diurnal cycle could be detected in two collocated MWR time series,
highlighting the value of ground-based radiometric measurements to monitor the short-term dynamics and photochemistry in the middle atmosphere. The quantification of these variations is limited by the rapidly increasing measurement noise, however some enhancements of the diurnal cycle are clearly visible in the upper stratosphere during wintertime, where the diurnal cycle is otherwise very small. Compared to the averaged monthly diurnal cycle, we find enhancement of $4-5$ times the monthly mean diurnal cycle amplitude lasting for a few days. In fact, the observed short-term variability of the ozone diurnal cycle
seems much higher than its intra-seasonal (month-to-month) or inter-annual variability during wintertime.

Regional (longitudinal) variability of the stratospheric ozone diurnal cycle has previously been identified by Schanz et al. (2014) in a model-based study from WACCM. They attributed the regional variability to changes in temperature (atmospheric tides), $O_x$ and $NO_y$. Our study supports that, in some cases, short-term variability in the ozone diurnal cycle can be attributed to changes in $NO_x$ concentrations through dynamical transport. In other cases, other processes might be acting to modify the
amplitude of the ozone diurnal cycle, for instance changes in atmospheric tides. Our study also shows that a CTM like BASCOE is able to simulate the changes of the ozone diurnal cycle amplitude due to changes in $NO_x$. In view of its significance, we believe that the reasons and importance of the short-term variability of the diurnal cycle should be further investigated with BASCOE over the globe.



It is beyond the scope of this publication to provide comprehensive analysis of this phenomenon but we aim at bringing
some new data to better understand stratospheric ozone diurnal cycle variability. It seems to be of particular interest in views
of the recent studies aiming at better accounting for the stratospheric ozone diurnal variability in satellite datasets (e.g. Frith
et al., 2020; Strode et al., 2022; Natarajan et al., 2023). Note that we focused our analysis on the upper stratosphere, where the
short-term variability was most visible in our observations, but short-term variations are not limited to this region. In fact, our
observations indicate that the variability is also present in the mesosphere and the lower stratosphere, where the role of $NO_x$ is
less important and where other processes likely dominate. To conclude, more work is definitely needed to assess the importance
of the short-term variability of the ozone diurnal cycle and confirm the potential role of other mechanisms influencing it.

*Code and data availability.* The GROMOS and SOMORA level 2 data are available from the Bern Open Repository and Information System
in the form of yearly netCDF files (Sauvageat et al., 2022a; Maillard Barras et al., 2022b).The recently harmonized calibration and retrieval
routines are freely available at https://doi.org/10.5281/zenodo.6799356. The data and analysis code reproducing all the results presented in
this manuscript are freely available. MLS v5 data (Schwartz et al., 2020) are available from the NASA Goddard Space Flight Center Earth
Sciences Data and Information Services Center (GES DISC): https://disc.gsfc.nasa.gov/). The ERA5 dataset (Hersbach et al., 2020) was
downloaded from the Copernicus Climate Change Service (C3S) Climate Data Store.





## Appendix A:  Monthly ozone profile comparisons

### A1    Additional plots for spring and autumn

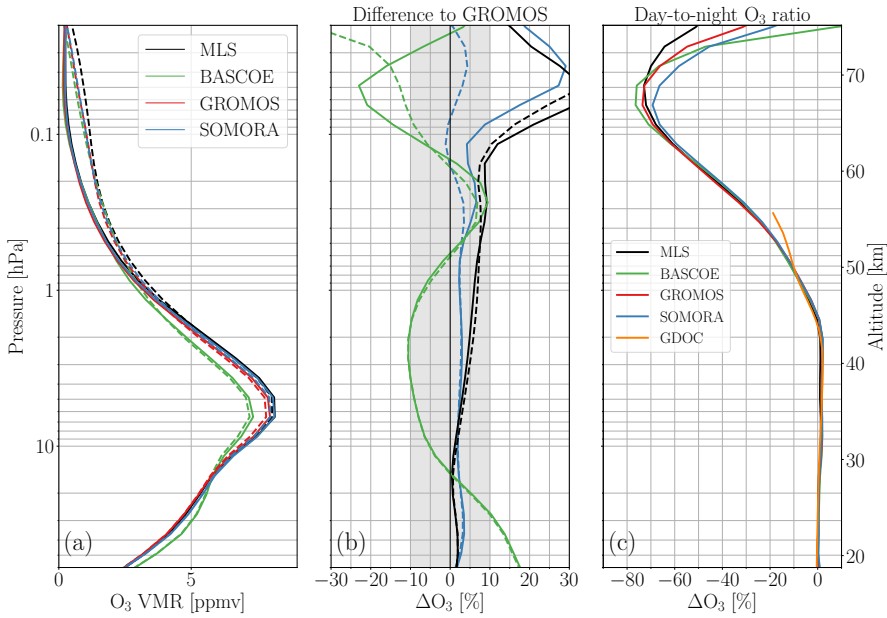

**Figure A1.** Same as Fig. 2 but for March.





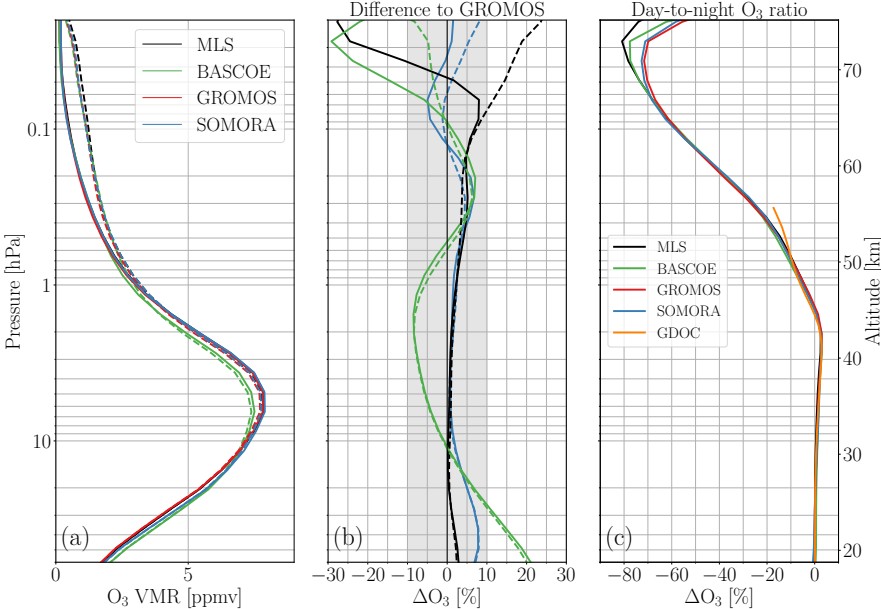

**Figure A2.** Same as Fig. 2 but for September.



# Appendix B: Monthly diurnal cycle

## B1   Additional plots for spring and autumn

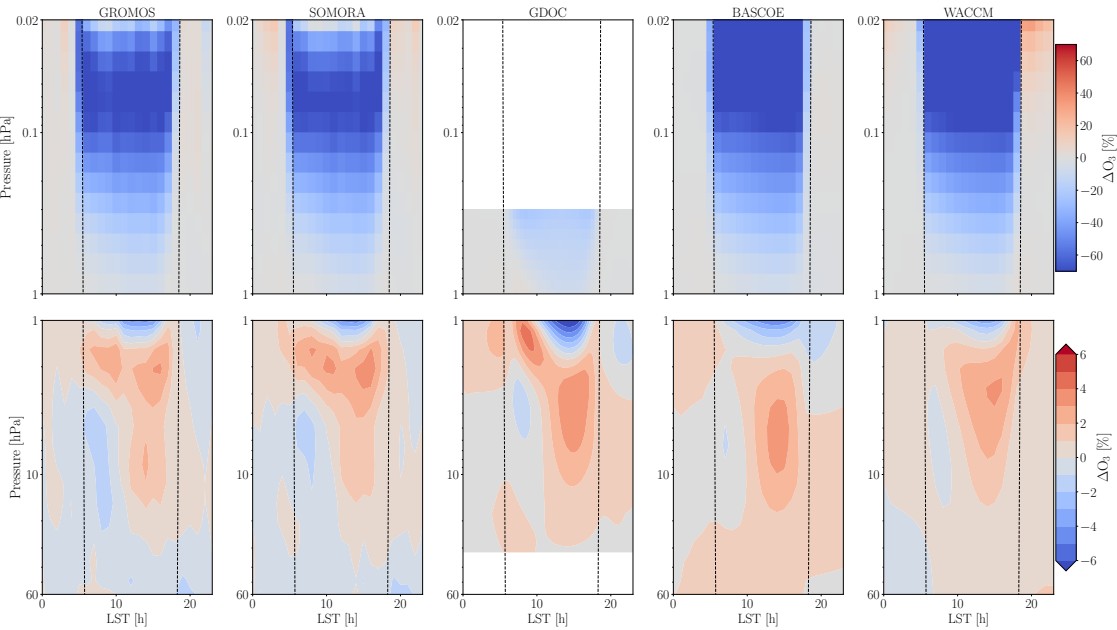

**Figure B1.** Same as Fig. 4 but for March.





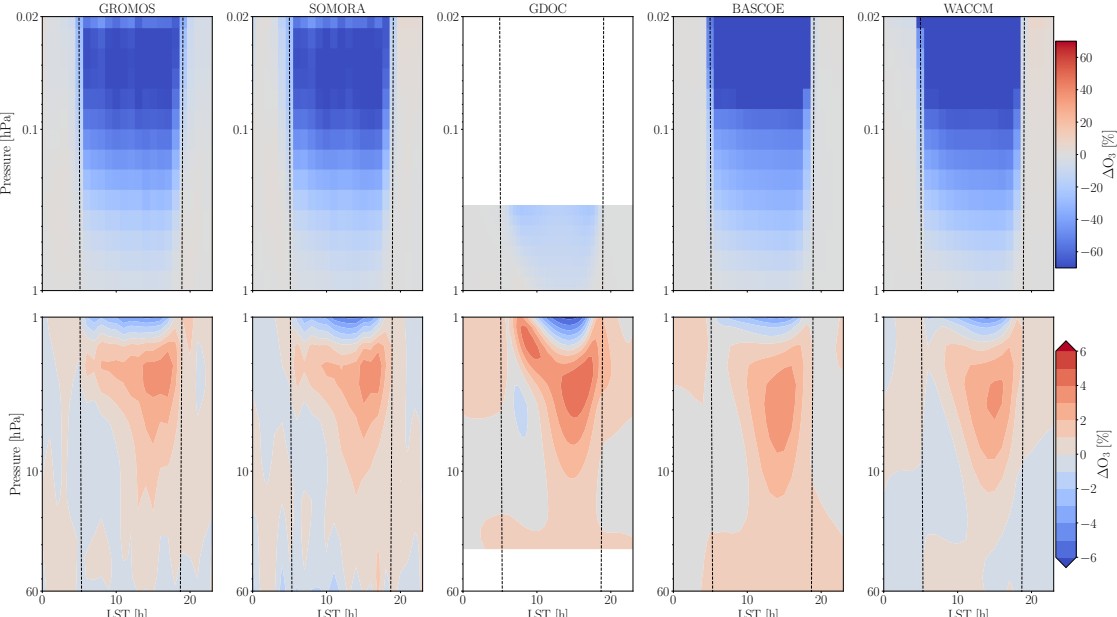

**Figure B2.** Same as Fig. 4 but for September.







**Figure B3.** Same as Fig. 6 but for March.





**Figure B4.** Same as Fig. 6 but for September.





## Appendix C: Short-term variability

### C1  Short-term

In this section, we show measurements and simulations of the short-term variability during the boreal winter 2016-2017.
Whereas more work is needed to unravel the complete picture of this winter, it shows another example of diurnal cycle enhancement associated with a sharp decrease of nitrogen oxides in the upper stratosphere.

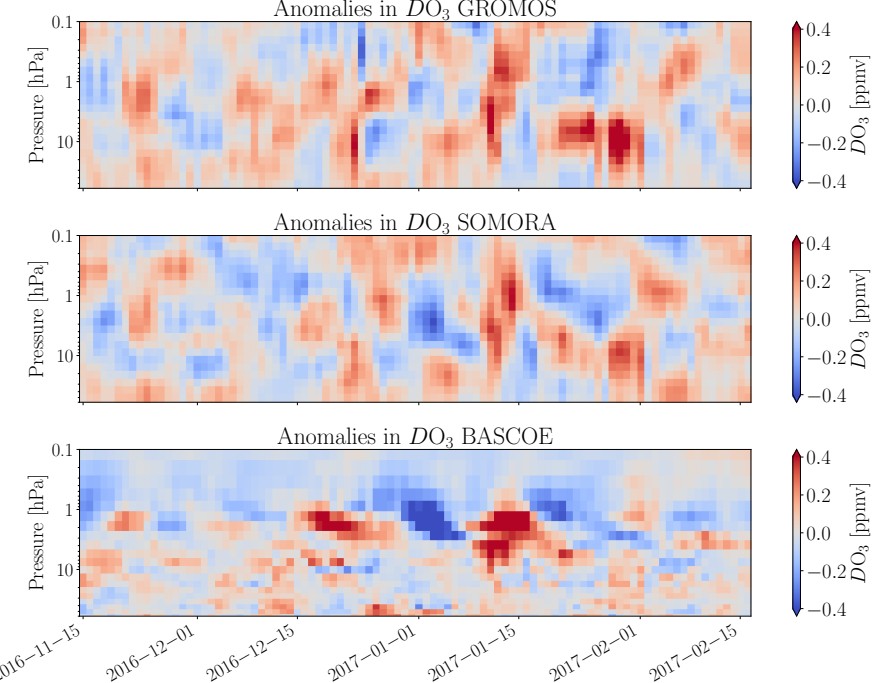

**Figure C1.** Anomalies in day-night $D_{O_3}$ from GROMOS, SOMORA and BASCOE during the boreal winter 2016-2017. For each dataset, we show the differences of $D_{O_3}$ compared to a monthly climatology computed on the decade 2010-2020.





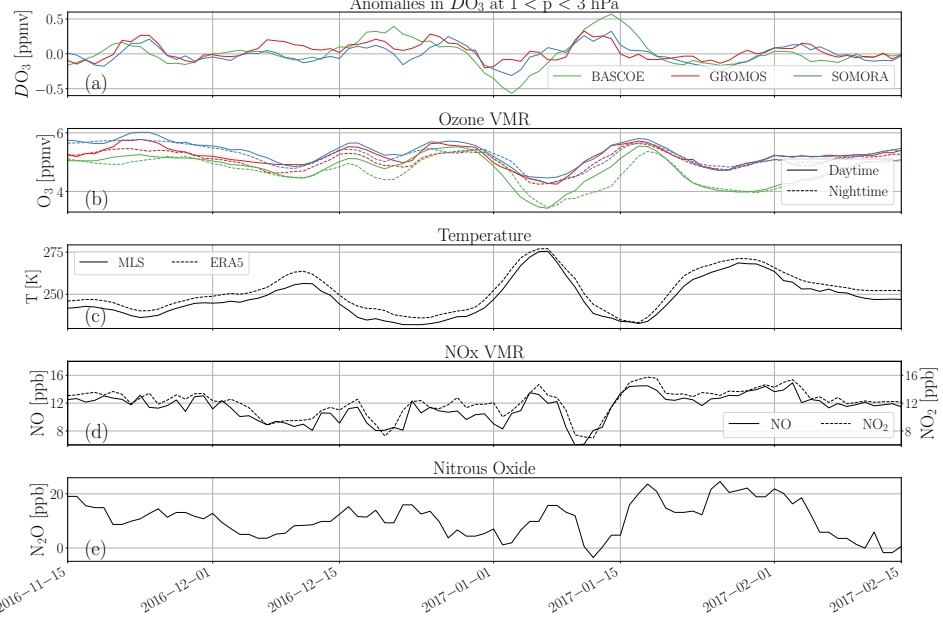

**Figure C2.** Time series of different quantities during the winter 2016-2017, all averaged between 3 and 0.8 hPa. (a) is the $D_{O_3}$ anomalies, (b) shows the ozone VMR of the three dataset during daytime and nighttime, (c) is temperature from MLS and ERA5, (d) are NO and $NO_2$ as simulated by BASCOE and (e) is $N_2O$ measurements from MLS.

*Author contributions.* ES carried out the data analysis, analysed the results, and prepared the manuscript. KH conceived the project and contributed to the interpretation of the results. EM provided the SOMORA data and contributed to the interpretation of the results. SH performed a preliminary study within the frame of his master thesis. QE performed the BASCOE simulation and provided valuable help on the interpretation of all model results. AH and AM contributed to the interpretation of the results. All of the authors discussed the scientific findings and provided valuable feedback for the manuscript editing.

*Competing interests.* The authors declare that they have no conflict of interest.

*Acknowledgements.* This work has been partly funded by MeteoSwiss and the Swiss Global Atmospheric Watch program. The authors would like to acknowledge all the people involved in the design and operation of GROMOS and SOMORA. Also, they would like to thank the developers of the Atmospheric Radiative Transfer Simulator (Buehler et al., 2018), version 2.4 and their precious support to setup the ozone retrievals. In addition, we thank the numerous contributors to the free and open source software packages used for the data analysis, in particular xarray, matplotlib, Typhon and pyretrieval.





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
