# Peer review of "Microwave radiometer observations of the ozone diurnal cycle and its short-term variability over Switzerland"

_EGUsphere, 2023_

## Author Comment (AC3)

**Response to Anonymous Referee #2 (RC2):**

We would like to thank the second referee for her/his time, positive feedbacks and valuable comments. Please find below the original comments and the authors' response (in blue). Except where mentioned, figures and line numbers refer to the original submitted manuscript.

##############################################################################
##############################################################################

This paper uses the harmonized timeseries from two ground-based microwave radiometers along with output from several models to investigate the diurnal cycle of ozone in the stratosphere and mesosphere over Switzerland. An interesting finding of this paper is that the amplitude of the ozone diurnal cycle has short-term variability. The paper is well-organized and provides useful data on the diurnal cycle of ozone. I list some general and specific comments below.

**General comments:**

Section 3.3: Tides are mentioned in the abstract and conclusion but could be discussed here as well to more strongly tie this section to the abstract and conclusions. It might also be helpful to show the timeseries for one of the observations (similar to Fig. 8a) but with the daily mean overplotted, to help distinguish changes in the diurnal cycle from changes in daily mean ozone, since both seem to be happening in the time series.

The authors agree that a discussion on tides effect on the ozone diurnal cycle could be added in this section and we propose to add a dedicated paragraph at the end of section 3.3 in a revised version of our manuscript. Regarding the suggestion of the referee to show the daily mean time series of ozone during one of the observation, we would argue that this is shown already in Fig. 10(b). If added to Fig. 8(a), we believe that it slightly degrades the readability of the figure (see Figure 1 below). If the referee agrees, our suggestion would be to keep Fig. 8(a) without the daily mean over-plotted lines and add some daily grid lines to highlight the time scale of the ozone changes.

[Figure]

*Figure 1: Reproduction of Fig. 8 with daily mean overplots (dash lines)*

**Specific comments:**

Line 125: Are the day and night averaging kernels very different, and if so, how does this affect the results?

No, the day and night AVKs are very similar for the two instruments, which is why we showed only the daily averaged ones in our manuscripts. In Figure 2 below, the reviewer will find a comparison plot of the averaged day and night AVKs for GROMOS and SOMORA:

Daytime

[Figure]

Nighttime

[Figure]

*Figure 2: Daytime (top) and Nighttime (bottom) averaging kernels (AVKs) for GROMOS (a) and SOMORA (b)*

Therefore, we consider that the difference between day and nighttime AVKs will not affect significantly the results for the diurnal cycle. In fact, because of their sensitivity to the weather conditions (through noise level and signal-to-noise ratio of the measurements), the AVKs show a seasonal variability which could influence the monthly diurnal cycle. As the ozone diurnal cycle itself shows a strong seasonal variability this effect is very difficult to quantify though. In any case, it will not affect the monthly comparisons against the various model dataset, nor the cross comparisons between the two radiometers.

####################

Line 151: Is it because the profile is normalized that you cannot apply the kernel, or because it is a monthly average?

Indeed, the GEOS-GMI Diurnal Ozone Climatology (GDOC) climatology only contains the hourly ozone values normalized to ozone midnight values ($\Delta O_3$). To apply the AVK smoothing procedure we would need the "original" ozone profiles used to produce the GDOC (see eq. (1) from our original manuscript).

We decided not to ask for the GDOC raw ozone profiles because we would then need to recompute the zonally averaged diurnal cycle after the AVK smoothing. Therefore, it could not be considered to be a comparison against the original climatology anymore. Also, we would argue that the effect of the smoothing can be seen with the other model datasets and would affect similarly the GDOC.

####################

Line 168: Why was a free-running perpetual year simulation used for this study instead of nudging?

Because it is often considered that the ozone diurnal cycle has relatively low (however not negligible) inter-annual variability. Therefore, we thought that it would be interesting to compare this free-running simulation against the BASCOE chemistry transport model (CTM) and the two radiometers. The fact that all of them agree well tends to confirm that the inter-annual variability of the diurnal might indeed be small, not only compared to seasonal variations but also against short-term variability.

####################

Section 2.2.2: Please provide some information on the chemical mechanism in BASCOE

Agreed, we will modify section 2.2.2 to add a more thorough description of the BASCOE CTM.

####################

Line 261: This might be easier to see if there was a plot of just the magnitude of the diurnal cycles.

We agree with the referee that an overview plots of the diurnal cycle amplitude was actually missing in our submitted manuscript. It is quite difficult to represent at once the whole middle atmospheric diurnal cycle so we would suggest to add an overview plots focusing on the stratosphere, where larger biases are observed. An example of such a plot is shown in Figure 3 below. The authors would add this figure at the beginning of Section 3.2 where we discuss the monthly ozone diurnal cycle and use it to introduce the monthly diurnal cycle and discuss the upper stratospheric bias from the GDOC with respect to the four seasons.

[Figure]

*Figure 3: Diurnal cycle amplitude in the stratosphere shown as the percentage point difference between the maximum and the minimum values of ΔO₃ (as defined in the original manuscrit) for four different months as proxis for the four seasons.*

---

## Author Response (AR1)

**Author's final response**

We would like to thank again the referees for their time, positive feedbacks and valuable comments. The point-to-point responses provided during the interactive discussion on EGUsphere are reproduced below for the two referees and some parts have been updated together to fit with the revised manuscript. Please find below the original comments and the authors' response (in blue). Note that figure and line numbers refers to the original submitted manuscript.

###############################################################################
###############################################################################

**Response to Anonymous Referee #1 (RC1):**

**General comments:**

This study provides new results on the diurnal variability of ozone; the data figures are very good based on the two nearly co-located ground-based microwave radiometer instruments. The authors emphasize monthly and seasonal diurnal ozone scaling factors, but they also report finding significant sub-monthly diurnal ozone variations during northern hemisphere winter. Even so, the original stated goal of the study is to generate a refined diurnal model for the purpose of merging multiple datasets for analyses of long-term ozone time series and for comparisons with model ozone time series. Thus, I would argue that they are showing that it is best to avoid winter hemisphere data for that purpose. Winter anomalies in temperature may be more important than those of NOx, but it is not easy to assess that prospect because of the low vertical resolution of the MW ozone profiles.

The authors agree completely with the general comment, especially with the part about the assessement of temperature effect on the ozone diurnal cycle anomalies. In fact, our study aims at showing that the sub-monthly variability can be observed using microwave radiometers, thus confirming some modeling studies. By focusing on a single case study, we wanted to show that chemisty could have an impact on the winter anomalies but we are not able to rule out the effect of temperature. Also, we believe that the sub-monthly variability can have multiple origins and that in some cases, the temperature changes will clearly prevails over the composition changes (e.g. NOx). We are currently attempting to correlate our ozone diurnal cycle anomalies with collocated temperature measurements but as the referee rightly mentioned, it is quite challenging because of the low vertical resolution and the low signal-to-noise ratio of the radiometers.

###############################################################################
###############################################################################

**Specific comments:**

Line 73—It would be helpful to learn at this point why there was an overestimate of the ozone diurnal cycle previously. Also define GROMOS here.

The exact source of the previous overestimation of the ozone diurnal cycle is difficult to identify but was due either to the calibration or retrieval algorithms. In fact, with the old retrieval algorithm, GROMOS was significantly less sensitive to ozone changes above approx. 45-50km which

probably explained most of the discrepancies between the old GROMOS series and the modelled ozone diurnal cycle.

We have now defined GROMOS in the introduction directly.

######################

Line 134—Here the authors give two specific overpass times for the MLS measurements, while on line 179 they indicate a more general range of time. Which is correct?

The overpass times for MLS indicated in Line 134 were the correct ones. The times indicated previously in Line 179 are the time ranges we used for the measurements and model datasets, which were chosen close to the MLS overpass times. We have now clarified the choice of time taken for the models and the observations on Line 195 of the revised manuscript.

######################

Line 213—What is the source of the noisy appearance? Gravity waves, perhaps?

In the context of line 213, the noise source of the daily diurnal cycle is due to the rather low signal-to-noise ratio of the radiometers themselves. The computation of the ozone diurnal cycle divides 2 hourly ozone profiles which are noisy by essence, resulting in a high noise level, so that this is not possible to use a single day to get an accurate view of the diurnal cycle. Averaging over multiple days reduces the noise and unravel the diurnal patterns of ozone in the middle atmosphere.

Regarding gravity waves, it is a very interesting point as gravity wave-induced ozone changes are of particular interest to our group. We believe that they can impact as well the ozone diurnal cycle by adding additional noise to the measurements when considering only a single day, however, this additional noise likely remains small compared to the inherent noise of the radiometer itself. As mentioned in the general comment's answer, we are currently looking for potential ozone response to gravity waves in the middle atmosphere but it remains challenging.

**Response to Anonymous Referee #2 (RC2):**

This paper uses the harmonized timeseries from two ground-based microwave radiometers along with output from several models to investigate the diurnal cycle of ozone in the stratosphere and mesosphere over Switzerland. An interesting finding of this paper is that the amplitude of the ozone diurnal cycle has short-term variability. The paper is well-organized and provides useful data on the diurnal cycle of ozone. I list some general and specific comments below.

**General comments:**

Section 3.3: Tides are mentioned in the abstract and conclusion but could be discussed here as well to more strongly tie this section to the abstract and conclusions. It might also be helpful to show the timeseries for one of the observations (similar to Fig. 8a) but with the daily mean overplotted, to help distinguish changes in the diurnal cycle from changes in daily mean ozone, since both seem to be happening in the time series.

[Figure]

*Figure 1: Reproduction of Fig. 8 with daily mean overplots (dash lines)*

The authors agree that a discussion on tides effect on the ozone diurnal cycle could be added in this section and we now added a dedicated paragraph at the end of section 3.3 in the revised version of our manuscript. Regarding the suggestion of the referee to show the daily mean time series of ozone during one of the observation, we would argue that this is shown already in Fig. 11(b) of the revised manuscript. If added to Figure 1, we believe that it slightly degrades the readability of the figure. If the referee agrees, our suggestion would be to keep Fig. 9(a) from the revised manuscript without the daily mean over-plotted lines and add some daily grid lines to highlight the time scale of the ozone changes.

**Specific comments:**

Line 125: Are the day and night averaging kernels very different, and if so, how does this affect the results?

No, the day and night AVKs are very similar for the two instruments, which is why we showed only the daily averaged ones in our manuscripts. In Figure 2 below, the reviewer will find a comparison plot of the averaged day and night AVKs for GROMOS and SOMORA:

[Figure]

*Figure 2: Daytime (top) and Nighttime (bottom) averaging kernels (AVKs) for GROMOS (a) and SOMORA (b)*

Therefore, we consider that the difference between day and nighttime AVKs will not affect significantly the results for the diurnal cycle. In fact, because of their sensitivity to the weather conditions (through noise level and signal-to-noise ratio of the measurements), the AVKs show a seasonal variability which could influence the monthly diurnal cycle. As the ozone diurnal cycle itself shows a strong seasonal variability this effect is very difficult to quantify though. In any case, it will not affect the monthly comparisons against the various model dataset, nor the cross comparisons between the two radiometers.

######################

Line 151: Is it because the profile is normalized that you cannot apply the kernel, or because it is a monthly average?

Indeed, the GEOS-GMI Diurnal Ozone Climatology (GDOC) climatology only contains the hourly ozone values normalized to ozone midnight values ($\Delta O_3$). To apply the AVK smoothing procedure we would need the "original" ozone profiles used to produce the GDOC (see eq. (1) from our original manuscript).

We decided not to ask for the GDOC raw ozone profiles because we would then need to recompute the zonally averaged diurnal cycle after the AVK smoothing. Therefore, it could not be considered to be a comparison against the original climatology anymore. Also, we would argue that the effect of the smoothing can be seen with the other model datasets and would affect similarly the GDOC.

######################

Line 168: Why was a free-running perpetual year simulation used for this study instead of nudging?

Because it is often considered that the ozone diurnal cycle has relatively low (however not negligible) inter-annual variability. Therefore, we thought that it would be interesting to compare this free-running simulation against the BASCOE chemistry transport model (CTM) and the two radiometers. The fact that all of them agree well tends to confirm that the inter-annual variability of the diurnal might indeed be small, not only compared to seasonal variations but also against short-term variability.

######################

Section 2.2.2: Please provide some information on the chemical mechanism in BASCOE

Agreed, we modified section 2.2.2 to add a more thorough description of the BASCOE CTM.

######################

Line 261: This might be easier to see if there was a plot of just the magnitude of the diurnal cycles.

We agree with the referee that an overview plots of the diurnal cycle amplitude was actually missing in our submitted manuscript. It is quite difficult to represent at once the whole middle atmospheric diurnal cycle so we would suggest to add an overview plots focusing on the stratosphere, where larger biases are observed. An example of such a plot is shown in Figure 3 below. The authors added this figure at the end of Section 3.2 where we discuss the monthly ozone diurnal cycle and use it to discuss the upper stratospheric bias from the GDOC in winter and the differences between the different months.

[Figure]

*Figure 3: Diurnal cycle amplitude in the stratosphere shown as the percentage change between the maximum and the minimum values of ΔO3 (as defined in the manuscrit) for all months.*

---

## Author Response (AR2)

Dear Editor,

Thank you for the careful handling of our manuscript and for your technical corrections.

We have now addressed all your last comments except for the one on P4, L106, because the abbreviation IAP is defined on L103 already. On P9, L237, we have reorganised the two sentences to make them clearer.
For the spell check, we have asked a native colleague so we hope that everything is now correct.

Best regards,
Eric Sauvageat on behalf of the authors.